# The Role of Nitrate Supply in Bioactive Compound Synthesis and Antioxidant Activity in the Cultivation of *Porphyra linearis* (Rhodophyta, Bangiales) for Future Cosmeceutical and Bioremediation Applications

**DOI:** 10.3390/md22050222

**Published:** 2024-05-15

**Authors:** Débora Tomazi Pereira, Nathalie Korbee, Julia Vega, Félix L. Figueroa

**Affiliations:** Experimental Center Grice Hutchinson, University Institute of Blue Biotechnology and Development (IBYDA), University of Malaga, Lomas de San Julián, 2, 29004 Málaga, Spain; debora.tomazi@uma.es (D.T.P.); nkorbee@uma.es (N.K.); juliavega@uma.es (J.V.)

**Keywords:** antioxidant activity, bioremediation, cosmetic, macroalgae, mycosporine-like amino acids, nitrate assimilation

## Abstract

*Porphyra sensu lato* has economic importance for food and pharmaceutical industries due to its significant physiological activities resulting from its bioactive compounds (BACs). This study aimed to determine the optimal nitrate dosage required in short-term cultivation to achieve substantial BAC production. A nitrate experiment using varied concentrations (0 to 6.5 mM) revealed optimal nitrate uptake at 0.5 mM in the first two days and at 3 and 5 mM in the last five days. Polyphenols and carbohydrates showed no differences between treatments, while soluble proteins peaked at 1.5 and 3 mM. Total mycosporine-like amino acids (MAAs) were highest in algae incubated at 5 and 6.5 mM, and the highest antioxidant activity was observed in the 5 mM, potentially related to the MAAs amount. Total carbon and sulfur did not differ between treatments, while nitrogen decreased at higher nitrate. This discovery highlights the nuanced role of nitrate in algal physiology, suggesting that biological and chemical responses to nitrate supplementation can optimize an organism’s health and its commercially significant bioactive potential. Furthermore, given its ability to absorb high doses of nitrate, this alga can be cultivated in eutrophic zones or even in out-/indoor tanks, becoming an excellent option for integrated multi-trophic aquaculture (IMTA) and bioremediation.

## 1. Introduction

Algae cultivation is known as aquaculture and started in Japan more than 200 years ago with the red macroalgae *Porphyra* [1]. However, there are studies that report the cultivation of *Porphyra* in 960–1279 in China [2]. *Porphyra sensu lato* species grow on all continents, but the largest cultivation is in Asian countries (approximately 99%); however, countries in Europe (Norway), Africa, and North America are beginning to produce this alga [3,4] since *Porphyra* is one of the most important economic, cultured, and consumed red algae in the world.

*Porphyra sensu lato* species are commonly known as Nori and have relevant economic importance in the food industry (mainly in Asian countries) as they are a great source of proteins, minerals, and vitamins [2]. In addition, this alga is also important for the pharmaceutical industry because it has important physiological activities due to its bioactive compounds (BACs), the majority of them have a nitrogenous composition, such as biliproteins, mycosporine-like aminoacids (MAAs), and proteins, and other non-nitrogenous BACs include polysaccharides and polyphenols. All these BACs have a positive effect on human health, including antiviral, anti-cancer, anticoagulant, antioxidant activity, regulation of the immune system regulation, and photoprotective action [5,6,7,8]. Among these BACs, MAAs attract attention because they are water-soluble, low molecular weight secondary metabolites that absorb UV radiation [9,10]. The main function of MAAs is their photoprotective ability, described as “microbial sunscreens” [11,12,13]. Additionally, MAAs have antioxidant properties, and they have been studied for their anti-cancer, anti-photoaging, wound healing, and anti-inflammatory effects [14,15,16]; currently, MAAs are also being investigated as a new generation of eco-friendly sunscreens [17,18,19].

Integrated multi-trophic aquaculture (IMTA) is a modern farming technology in aquaculture, where nitrogen is one of the primary effluents released [20]. IMTA eliminates the need for extensive use of chemical nitrogen fertilizers, a common practice in terrestrial crop production that significantly increases total production costs and environmental contamination. Additionally, IMTA presents numerous advantages, including guaranteed availability of biomass, improvement in its quality, and a consistent chemical composition [21,22]. These aspects contribute to more sustainable aquaculture practices and facilitate the production of high-value nitrogen compounds, like the BACs found in *Porphyra*, which have wide-ranging applications.

Moreover, bioremediation is a technology that employs plants and algae to eliminate or neutralize pollutants in the environment, thereby rendering them harmless [23]. Generally, studies on bioremediation involve *Ulva* sp., a green alga, [24] and *Laminaria* sp., a brown alga, [25,26], which, however, have a lower market value compared to *Porphyra*. It is known that *Porphyra* can absorb dissolved nitrogen [20,27,28,29]; however, it is always analyzed with low dissolved concentrations. Furthermore, in marine environments, nitrogen is a crucial limiting factor for algal growth [30,31]; therefore, developing the cultivation of high-valued seaweed species such as *Porphyra* to reduce eutrophication and improve the quality of coastal waters would be of significant economic and environmental importance.

Even with all the benefits that the red macroalga *Porphyra* can provide, the main production occurs in farms in Asia at sea or in tanks, including two stages: the first one involves handling the microscopic/conchocelis phase in the laboratory, and the second one involves ‘seeding’ these spores on nets on fixed or semi-floating rafts in the open sea, to develop the macroscopic phase, without much control over abiotic factors [32]. These farms focus on biomass production for the food industry without assessing and quantifying the BACs. According to Future Market Insights, the algae products market was USD 5.41 billion in 2023, and this market is expected to achieve USD 9.15 billion (CAGR 5.41%) by 2033. Thus, this research was carried out with the goal of cultivating *Porphyra linearis* Greville to determine the optimal nitrate dosage required in the water to yield substantial production of BACs, considering the IMTA, bioremediation, and the expansive production of this important macroalgae.

## 2. Results

### 2.1. Nitrate Quantification: Nitrate Uptake Efficiency (NUE) and Nitrate Uptake Rate (NUR)

Considering the nitrate uptake efficiency, samples exposed to 0.15 and 0.5 mM demonstrated the highest nitrate efficiency absorption over two days, whereas samples exposed to 3, 5, and 6.5 mM exhibited the lowest efficiency. By the conclusion of the last five days, the observed pattern from the two-day mark recurred, with higher concentrations showing reduced absorptions. In terms of the daily nitrate uptake rate, for two days of exposure, the highest rate of absorption was detected in 0.5 mM, and the lowest rate of absorption was in algae exposed to 6.5 mM. All other treatments showed median nitrate absorption, exhibiting statistical similarity with both the highest and the lowest. Considering the last five days of exposure, the daily nitrate uptake rate was highest in samples from 3 to 5 mM, followed in descending order with statistical differences from 6.5, 1.5, 0.5, and 0.15 mM (Table 1). A positive and strong Pearson correlation between nitrate quantity and nitrate uptake rate was observed in the last five days of exposure (r = 0.8440, *p* = 0.001, *n* = 21), but a negative correlation between nitrate concentration and nitrate uptake efficiency was present in the first two days (r = −0.4518, *p* = 0.040, *n* = 21).

### 2.2. Effects of Nitrate Concentrations on Polyphenols, Soluble Proteins, and Carbohydrates

A significant reduction in polyphenols, soluble proteins, and carbohydrates was observed during the nutrient-free acclimatization period when compared to field samples. When analyzed at seven days of exposure, the acclimatized samples exhibited the lowest polyphenol concentration, whereas all treated samples showed higher contents, although without significant statistical differences among them (Figure 1a). Regarding the soluble protein content of *P. linearis* after seven days of culture, all samples showed increased soluble protein content compared to the acclimatized sample, with 1.5 and 3 mM nitrate yielding the best response and 5 mM the least favorable (Figure 1b). In the case of carbohydrates, the acclimatized samples continued to show the lowest carbohydrate content in comparison to all treatments. Finally, after seven days of exposure, all nitrate concentrations displayed no significant statistical differences among them (Figure 1c).

Additionally, no significant Pearson correlation was observed between nitrate quantity and polyphenol, soluble protein, or carbohydrate contents on day seven of the experiment.

### 2.3. Effects of Nitrate Concentrations on Mycosporine-like Amino Acids (MAAs) Profiles

Three types of mycosporine-like amino acids (MAAs) were identified by HPLC: asterina-330, shinorine, and porphyra-334. When summing all MAAs detected (total MAAs), a statistically significant decrease was noted in the samples acclimatized without nutrients when compared to field samples, and there was an increase in total MAA content for samples exposed to 5 and 6.5 mM nitrate, with these samples exhibiting the highest total MAA content across all concentrations after seven days of exposure (Figure 2a).

For asterina-330, a statistically significant increase was noted in the samples acclimatized without nutrients when compared to field samples. In addition, no significant statistical differences were observed among all samples, including the acclimatized sample, after seven days of exposure (Figure 2b). When converting the percentage of asterina-330 relative to the total sum for each treatment, it was observed that all treatments showed higher amounts of this type of MAA compared to acclimatized samples (Figure 2b). Regarding shinorine, there was a statistically significant decrease in samples acclimatized without nutrients when compared to field samples. Shinorine was present only in the 0.15 mM nitrate sample at seven days, and no statistical difference was found between this treatment and the acclimatized sample, neither in concentration nor in relative percentage (Figure 2c). Considering porphyra-334, the most abundant MAA detected, samples exposed to 5 and 6.5 mM nitrate displayed the highest concentrations, whereas samples exposed to other nitrate concentrations, including the acclimatized sample, had the lowest content of this MAA type. When converting the percentage of porphyra-334 relative to the total sum, it was observed that all samples did not show statistical differences between them (Figure 2d).

Additionally, a positive and strong Pearson correlation between nitrate quantity and total MAAs content was observed at seven days (r = 0.6697, *p* = 0.001, *n* = 21).

### 2.4. Effects of Nitrate Concentrations on Antioxidant Activity (ABTS Test)

Regarding antioxidant activity, a statistically significant reduction was observed in the acclimatized samples after 14 days without nutrients, compared to the field samples. On the other hand, considering seven days of the experiment, there was an increase in antioxidant activity in all samples when compared to the acclimatized sample, and the samples exposed to 0.15, 1.5, 3, 5, and 6.5 mM nitrate demonstrated the highest antioxidant activity (Figure 3).

A positive Pearson correlation between nitrate quantity and antioxidant activity was noted at seven days of exposure (r = 0.5065, *p* = 0.019, *n* = 21), as well as a positive Pearson correlation was noted between the number of antioxidant agents and total MAAs (r = 0.5398, *p* = 0.012, *n* = 21), while no correlation was observed between the number of antioxidant agents and the concentration of polyphenols.

### 2.5. Effects of Nitrate Concentrations on Total Carbon, Nitrogen and Sulfur

Considering total carbon content, a statistically significant reduction was observed in the acclimatized samples after a no-nutrient period compared to the field samples. After seven days of exposure to various nitrate concentrations, no statistical difference was observed among all samples, including the acclimatized (Figure 4a).

Regarding nitrogen content, a decrease in the concentration was also observed when comparing the acclimatized sample with the field sample. After seven days of nitrate exposure, samples from 0, 0.15, and 0.5 mM showed the highest quantities, while samples from 3, 5, and 6.5 mM had the lowest (Figure 4b).

Concerning sulfur content, the acclimatized samples again showed a statistically significant decrease when compared to the field sample. In contrast, with seven days of exposure, no statistical difference was observed among all samples, encompassing the acclimatized sample (Figure 4c).

Regarding the stoichiometries, the C:N ratio was higher in the 3, 5, and 6.5 mM samples, while all other samples did not show a distinct ratio among them. Concerning the C:S and N:S ratios, samples treated with 6.5 mM exhibited the highest ratios, followed by the 0 mM samples. For the three ratios analyzed, acclimatized samples showed higher ratios than field samples (Figure 4d–f).

Additionally, a negative Pearson correlation between nitrate quantity and total nitrogen contents was observed at seven days of exposure (r = −0.8430, *p* = 0.001, *n* = 21), whereas for carbon and sulfur, no correlations were observed.

### 2.6. Principal Component Analysis (PCA) of BACs from P. linearis Extracts with Different Time Periods and Nitrate Concentrations

The PC1 and PC2 explained 52.84% of the variability in the samples across different nitrate concentrations, and there is not a clear distinction in group formation, with an overlap observed among all groups (Figure 5), showing that increased nitrate concentration did not produce significant differences in the most of BACs.

## 3. Discussion

The use of different and elevated nitrate concentrations in the cultivation water of *P. linearis* was evaluated, and significant differences in the biochemical response were observed. Within just two days of exposure, algae incubated to 0.15 and 0.5 mM of nitrate consumed approximately 90% of the available nitrate with a high daily uptake rate of around 0.40 mM·g^−1^ DW·day^−1^, while at higher nitrate concentrations, only about 5% was absorbed, which equals the concentration in mM absorbed at the lower concentrations. In the subsequent five days of the experiment, and without any new addition of nitrate, the daily nitrate uptake rate was higher in algae exposed to medium and high nitrate concentrations since those at lower concentrations had nearly depleted the nitrate within the first two days, leading to a decreased rate by the end of the experiment. *Porphyra dioica* exposed from 0.025 to 0.3 mM [20] exhibited the same pattern observed in *P. linearis* in the present work, where lower nitrate concentrations showed higher NUE and lower NUR. A similar pattern was observed in the red algae cultivated under low and high nitrate levels under solar radiation [18]. *Porphyra dioica*, in [20], at the lowest nitrate concentration, exhibited a nitrogen content of approximately 1%, whereas at the highest nitrate concentration, it showed 5%. In contrast, the algae in the present study, even after a no-nutrient acclimation period, started with nearly 5% nitrogen content. However, even with a nitrogen percentage considered normal and sufficient for cellular metabolism, the algae continued to absorb nitrate, increasing the production of BACs.

For all BACs analyzed, including polyphenols, soluble proteins, carbohydrates, and MAAs, along with antioxidant activity and the content of carbon, nitrogen, and sulfur, a decrease was observed in acclimatized samples when compared to field samples. This is likely due to the fact that the light intensity in the natural environment is much higher, which favors the process of photosynthesis, enhancing the metabolism of the algae and, consequently, increasing the production of compounds.

Polyphenols are secondary metabolites typically associated with antioxidant properties and protection against oxidative stress [33]. In plants and algae cultivated under stress conditions, such as nitrogen deficiency, a redirection in carbon reserves from secondary metabolism towards biomass accumulation can occur [33]. This can explain the decrease in polyphenol production in the absence of nitrate that was observed in acclimatized samples, corroborating studies with *Pyropia acanthophora* var. *brasiliensis* [34]. Furthermore, even though the acclimatization period was conducted under the same conditions and density as the experiment, the acclimatization took place in tanks, and at times, the thalli became more shaded, and the reduction in absorbed light, combined with the absence of nutrients, may have contributed to decreasing the quantity of the compounds, such as polyphenols. Phenolic compounds are synthesized via the shikimate pathway, also known as the shikimic acid pathway, from which the aromatic amino acid phenylalanine (C_9_H_11_NO_2_) is produced, serving as the initial intermediate in the phenylpropanoid synthesis pathway. Through the action of the enzyme phenylalanine ammonia-lyase, phenylalanine is converted into cinnamic acid, which is hydroxylated to p-coumaric acid. This is then converted to p-coumaroyl-CoA, which is the substrate used for the synthesis of phenolic compounds [35]. Additionally, it is known that the main polyphenols present in *Porphyra sensu lato* include catechins, epicatechins, gallic acid, 4-hydroxybenzoic acid, epigallocatechin, epigallocatechin gallate, salicylic acid, rutin, and hesperidin [36,37,38,39], and none of these molecules contain nitrogen in their composition, even though they are derived from phenylalanine, which contains one nitrogen in its molecular formula. This nitrogen, forming the initial molecule in the phenolic synthesis pathway, can be recovered from other cellular components. Therefore, the absence and presence of nitrate in the water of the experiment period would not be expected to favor the production of these compounds, explaining why no statistical difference was observed between the treatments.

Proteins are macromolecules and are considered primary metabolites, composed of amino groups (-NH_2_) along with carbon, hydrogen, and oxygen. In addition to performing essential functions such as structural support, signaling for growth and cell division, and substance transport, they can also act as pigments (phycobiliproteins) and antioxidant enzymes (catalase, superoxide dismutase, and peroxidases) [40]. The acclimation period led to a decrease in the soluble protein content compared with field samples, beyond the absence of nutrients, to the reduction in irradiance causing lower photosynthesis rates, which in turn slowed down metabolism and consequently reduced the quantity of the compounds, as mentioned above. Even in the absence of nitrate, the concentration of soluble proteins increased, which may be associated with more light intensity and the degradation of other cellular structures to produce these defensive antioxidant compounds. On the other hand, at higher nitrate concentrations in the water, the soluble protein concentration remained stable, even showing a slight decrease, possibly reflecting optimal cultivation conditions that do not necessitate a highly developed antioxidant apparatus.

Carbohydrates are non-nitrogenous molecules, primary metabolites, that offer growth and structural support through components like cellulose, mannose, and xylose while also facilitating energy storage in the form of starch [41]. Beyond these neutral polysaccharides, algae are capable of producing sulfated polysaccharides, including porphyran, for algae of the *Porphyra sensu lato* genus, which confers resistance. Initial samples, both from the field and acclimatized, exhibited low concentrations of carbohydrates. This can be justified by the constant renewal of nutrients in their natural environment, negating the need to produce reserve materials. Furthermore, being in their natural and optimal environment and already adapted, they do not require physical protection through the thickening of cell walls. Upon being brought into laboratory conditions with a lack of nutrients, the algae were unable to produce these substances. After the experiment with different nitrate concentrations, the carbohydrate concentration increased, and no differences were observed between treatments. Since carbohydrates are non-nitrogenous compounds, nitrate does not act as a carbohydrate builder. However, it can benefit metabolism and favor other metabolic pathways. Even the 0 mM sample displayed significant carbohydrate levels, indicating that photosynthesis is occurring at a good/optimal efficient electron transport rate. On the other hand, field samples could be undergoing photoinhibition since the algae receive very high radiation intensities, while acclimation samples are photoinhibited since they have been shaded during this period. Without photosynthesis at good/optimal levels, the entire growth metabolism and, consequently, the production of carbon molecules is disadvantaged.

The MAAs are secondary metabolites, nitrogenous compounds known for their UV-absorbing properties [9,10]. It is known that the production of MAAs can occur via two routes: the shikimate pathway and the pentose phosphate pathway, with the first one being the more common [10]. For both MAA synthesis pathways, the final steps involve the transformation of 4-deoxygadusol to mycosporine-glycine and, from there, to all known types of MAAs (porphyra-334, asterina-330, shinorine, palythinol, palythine, usujirene, and taurine), which requires the presence of nitrogen. Thus, the presence of high nitrate concentrations can favor the increased production of these compounds, which are of great commercial interest, reaching concentrations of 28 mg·g^−1^, as observed in algae exposed to 5 mM of nitrate. This treatment also showed lower concentrations of soluble proteins, demonstrating that this concentration of nitrate available in the water favored the allocation of nitrogen to the metabolism of MAAs. This level is significantly high compared to those previously described for *Porphyra* by [42] and [43], which were 12 and 15 mg·g^−1^, respectively. Korbee et al. [44] showed that an increased presence of ammonium (0 to 0.3 mM) led to higher production of MAAs in *Porphyra leucosticta*, with levels rising from 6.99 mg·g^−1^ to 9.67 mg·g^−1^. It is also known that the presence of blue light in the same algae boosts the production of these compounds, reaching approximately 11 mg·g^−1^ [45]. Barufi et al. [46] demonstrated that in *Gracilaria tenuistipitata*, MAAs were increased by the combined effects of nitrate presence (0.1 and 0.5 mM) and ultraviolet radiation (UVR) exposure. Similarly, Peinado et al. [46] observed the same pattern in *Porphyra columbina*, where the interaction of ammonium concentrations (0 to 0.3 mM) and UVR enhanced MAA production. This suggests the potential for future studies to further increase MAA concentration through combined blue light and UVR exposure. In the present research, it was observed that the period in laboratory culture caused the amount of shinorine to decrease drastically. It is noted that the percentage of porphyra-334 does not change statistically with the treatments; therefore, there is a substitution of shinorine for asterina-330, as described by [47]. According to the synthesis pathway, asterina-330 is produced from shinorine via palythinol by the addition to H_2_ and then by the loss of CH_3_ or directly from shinorine by the loss of CO_2_ [48]. Therefore, in these thalli exposed to 0.15 mM, this transition has not yet been completed, but it is expected to occur within a few more days of cultivation. Additionally, in the current study, although there is a significant increase in the total concentration of MAAs, their low molecular weight, approximately 200 to 400 Da, does not seem to result in an increased nitrogen content.

The thalli from all treatments and the acclimatized sample maintained concentrations within the normal range of carbon content [49,50], indicating good metabolism in laboratory conditions. The nitrogen content showed a decrease by the seventh day of the experiment with the higher nitrate concentrations, corroborating the amount of soluble proteins in these thalli. The sulfur content remained the same for all samples and for the acclimatized sample by the seventh day. This could be associated with the maintenance of sulfated carbohydrate production in the cell wall, porphyran, as the thickness of the cell wall is primarily related to the presence of stress factors, largely light stress [34,51]. By the seventh day with nutrients, the algae did not exhibit this physical defense mechanism. Increasing the nitrate in seawater without increasing CO_2_ or bicarbonate supply and phosphorus changes the stoichiometry, i.e., [52] the ratio of 163C:22N:1P, and this can have negative effects on photosynthetic activity since the proportion of substrate for photosynthesis as the CO_2_. Furthermore, the nitrate ratio can break the dynamic interaction between carbon and nitrogen metabolism in the use of ATP and NADPH produced in the photochemical reactions of photosynthesis [53,54,55].

In summarizing the findings, the PCA analyses show that there is no clear distinction in group formation, with an overlap observed among all groups. An extended duration of the experiment might be required to observe distinctions among the treatments.

Finally, the antioxidant activity increased by the seventh day of exposure, with the most significant increases in the samples of 5 mM. This could be closely linked to the marked increase in MAAs, UV screen substances with antioxidant, anti-inflammatory, and anticollagenase capacities [29,56,57,58,59] with high interest in the cosmeceutical industry [17,19,59,60].

## 4. Materials and Methods

### 4.1. Biological Material

The gametophytes of *P. linearis* were collected on the rocky shore of Santa Cristina Beach (43°61′ N and 8°18′ W; 0.7 µM of NO_3_^−^ (M‘y Ocean Pro—Copernicus Marine Open Data’ platform)), Galicia, Spain, in May 2023, by ‘Porto Muiños’ company. Algal thalli were transported to the University Institute of Blue Biotechnology and Development (IBYDA) of the University of Málaga (Spain) in plastic containers containing seawater inside and in a thermal box. In the laboratory, the thalli were washed with diluted artificial seawater (produced through the dilution of sea salt from saline in Cadiz, Spain), and healthy portions were selected and followed for acclimation with artificial seawater (32 psu) without nutrients for 14 days under controlled conditions: photosynthetic active radiation (PAR) of 120 µmol photons m^−2^s^−1^ (White LED light, 5000 K, 54 W, NU-8416, Nuovo), temperature of 15  ±  2 °C, photoperiod of 12 h, and continuous aeration. After the acclimation period, samples, as well as field samples (a group referred to as initial samples), were frozen at −80 °C for metabolite extraction.

### 4.2. Conditions of Cultivation

The culture room conditions were the same as those in the acclimation period. Following acclimation, portions of *P. linearis* (3.0  ±  0.05 g) were cultured in UV transparent polyvinylmethacrylate (Plexiglas^®^ XT-29080, Darmstadt, Germany) cylindrical vessels filled with 750 mL of artificial seawater (32 psu), with the addition of nitrate in different concentrations (0, 0.15, 0.5, 1.5, 3, 5 and 6.5 mM of KNO_3_) (3 cylinders per each nitrate concentration) and 24 μM of PO_4_ (the same for all treatments). The experiment lasted 7 days. On the seventh day of the experiment, samples were frozen at −80 °C for extraction of the metabolites.

### 4.3. Nitrate Quantification: Nitrate Uptake Efficiency (NUE) and Nitrate Uptake Rate (NUR)

The nitrate quantification in water was carried out in samples taken from days 0, 2, and 7 of the experiment. A 1 mL sample was added with 50 μL of Griess-reagent (Sulphanilamide in HCl 1.3N (1% *w*/*v*) and N-(1-naphthyl)-ethylenediamine dihydrochloride (NED) in MiliQ water 0.005%, 1:1) and 100 μL VCl_3_-reagent (Vanadium (III) chloride in HCl 6N (2% *w*/*v*)) and gently mixed and incubated in a temperature controlled dry bath at 60 °C for 25 min. Then, the samples were cooled down to room temperature, and the absorbance was measured at 540 nm [61]. The quantification of nitrate total was determined using a standard curve of KNO_3_^−^ (1 to 25 µM—R^2^ = 0.99; y = 0.031567x; where y represents absorbance and x represents concentration).

### 4.4. Bioactive Compounds (BACs) Extraction

The extractions of BACs were performed at the IBYDA laboratories of Malaga University, Spain, with samples from the initial period (field and acclimatized samples) and after 7 days of incubation. In a test tube, 500 mg of fresh algae was mixed with 5 mL of extraction solvent (distilled water with 2.5% sodium carbonate (SC)), performing alkaline hydrolysis. After combining the algae with the extraction solvent, it was macerated for 30 s using an UltraTurrax^®^ (T25, IKA, Staufen, Germany) (18,000 rpm) and then subjected to extraction at 80 °C for 1.5 h in a dry bath. Following extraction, the samples were centrifuged at 2721.6× *g*, the supernatant was collected, and the pH was adjusted using lactic acid to achieve a pH of 7.0 (pHmeter Horiba LaquaTwin, Kyoto, Japan). Thus, the algal extracts were prepared for the measurement of concentrations of BACs (polyphenols, soluble proteins, and carbohydrates) and antioxidant activity.

### 4.5. Soluble Polyphenols

The analysis of polyphenol compounds was carried out using the spectrophotometric Folin–Ciocalteu method based on [62]. Aliquots of 100 µL from the diluted algal extracts were added to 700 µL distilled water, 50 µL of the Folin–Ciocalteu reagent (Sigma-Aldrich, Steinheim, Germany), and 150 µL of 20% sodium carbonate, and incubated for 2 h at 4 °C. Subsequently, readings were taken at 760 nm using a UV–visible spectrophotometer. The quantification of polyphenol compounds was determined using a standard curve of phloroglucinol (1 to 20 µg·mL^−1^—R^2^ = 0.99; y = 0.0586x; where y represents absorbance and x represents concentration). The analyses were performed in triplicate, and the results were expressed in mg of phloroglucinol per g of DW.

### 4.6. Soluble Proteins

The analysis of soluble proteins was carried out using the spectrophotometric Bradford method based on [63]. Aliquots of 50 µL from the diluted algal extracts were added to 750 µL phosphate buffer (0.1 M, pH 6.5) and 200 µL of the Bradford reagent (BioRad) and incubated for 15 min at room temperature. Subsequently, readings were taken at 595 nm using a UV–visible spectrophotometer. The quantification of soluble proteins was determined using a standard curve of bovine albumin (Sigma-Aldrich) (4 to 60 µg·mL^−1^—R^2^ = 0.99; y = 0.0244x; where y represents absorbance and x represents concentration). The analyses were performed in triplicate, and the results were expressed in mg of bovine albumin per g of DW.

### 4.7. Soluble Carbohydrates

The analysis of carbohydrates was carried out using the spectrophotometric method based on [64]. Aliquots of 500 µL from the diluted algal extracts were added to 1 mL anthrone 0.2% (*w*/*v*). The samples were subjected to a dry bath at 100 °C for 3 min to carry out the reaction and proceeded to be read on a UV–visible spectrophotometer at 630 nm. The quantification of carbohydrates was determined using a standard curve of glucose (25 to 200 µg·mL^−1^—R^2^ = 0.99; y = 0.01109x; where y represents absorbance and x represents concentration). The analyses were performed in triplicate, and the results were expressed in mg of glucose per g of DW.

### 4.8. Mycosporine-like Amino Acids (MAAs)

The extractions for the measurement of concentrations of mycosporine-like amino acids (MAAs) were performed with samples from the initial period (field and acclimatized samples) and 7 days of the experiment. In a test tube, 500 mg of fresh algae were mixed with 5 mL of extraction solvent (distilled water). After combining the algae with the extraction solvent, they were macerated for 30 s using an UltraTurrax^®^ (18,000 rpm) and then subjected to extraction at 80 °C for 1.5 h. Following extraction, the samples were centrifuged at 2721.6× *g*, and the supernatant was collected.

The determination of MAA concentrations was carried out according to [47] with modifications of [65]. One mL of algal extracts was filtered through a 0.2 µm membrane before chromatographic analysis using an HPLC system (1260 Agilent InfinityLab Series, Santa Clara, CA, USA) with a Diode-Array Detection (DAD) detector. The MAAs separation was performed injecting 10 µL of extract algal into a C8 Luna Column (250 mm length and 4.6 mm diameter; Phenomenex, Aschaffenburg, Germany) that was kept at 20 °C and the samples at 10 °C, using an isocratic run containing 1.5% aqueous methanol (*v*/*v*) plus 0.15% acetic acid (*v*/*v*) in Miliq water as the mobile phase, with a flow rate of 0.5 mL·min^−1^, and each run took 30 min. MAAs were detected at 320 and 330 nm. Isolated MAAs through HPCCC were used as standards [66]. The quantification was performed using published molar extinction coefficients (ε) of the different MAAs [67,68] and were expressed in mg·g^−1^ of DW.

### 4.9. Antioxidant Activity

Antioxidant capacity was determined by the ABTS radical scavenging assay. Firstly, the radical cation ABTS^+•^ was prepared by mixing 7 mM of ABTS (2,2′-azino-bis (3-ethylbenzothiazoline-6-sulphonic acid; Sigma-Aldrich) and 2.45 mM of potassium persulfate (K_2_S_2_O_8_) in a sodium phosphate buffer solution (0.1 M, pH 6.5). The mixture was incubated in darkness at room temperature for 16 h for the complete formation of the radical. For the reaction, the ABTS^+•^ was diluted with phosphate buffer until the absorbance at 727 nm was 0.75 ± 0.05. The assay was performed by adding 950 μL of the diluted ABTS^+•^ solution and 50 μL of diluted algal extract, according to [69]. The samples were shaken, and absorbance was recorded by a UV–visible spectrophotometer at 727 nm at the beginning of the reaction (DOi) and after 8 min of incubation (DOf). The percentage of antioxidant activity (AA%) was calculated according to the formula:AA% = [(absDOi − absDOf)/absDOi] × 100

The antioxidant compounds concentration was calculated using a standard curve of Trolox (6-hydroxy-2,5,7,8-tetramethylchroman-2-carboxylic acid), (Sigma-Aldrich) (20 to 100 µg/mL—R^2^ = 0.99; y = 15.8928x; where y represents absorbance and x represents concentration), and the results expressed in μmol of Trolox equivalent antioxidant capacity (TEAC)·g^−1^ of DW.

### 4.10. Total Carbon, Nitrogen, and Sulfur

Total carbon (C), nitrogen (N), and sulfur (S) contents of the dry algal biomass were evaluated using a LECO-932 CNHS elemental analyzer (St. Joseph, MI, USA) with samples from the initial period (field and acclimatized samples) and day 7 of the experiment, in the Research Support Central Services (SCAI, University of Malaga, Malaga, Spain). The analyses were performed in triplicate, and the results were expressed in %.

### 4.11. Statistical Analysis

The data passed the Shapiro–Wilk normality test and the Bartlett test for homogeneity of variances, and all samples were within the normal range and exhibited homoscedasticity. Afterward, the data were analyzed by unifactorial analysis of variance (ANOVA) (considering nitrate concentrations) followed by Tukey’s a posteriori test (*p* ≤ 0.05). Statistical analyses were performed using the Statistica software package (Release 10.0). Data were also tested by principal component analysis (PCA), and the graphical design was carried out using scripts written in Python language through the software Spyder to determine similarities of physiological variables that were analyzed in the present work.

## 5. Conclusions

It is concluded that *P. linearis* from Galicia, Spain, is an alga adapted to regions with high natural nitrate concentrations, capable of being cultivated in eutrophic zones or even in out- and indoor tanks with the addition of nitrate to the water, thereby possessing high levels of nitrogenous compounds which confer significant resistance in environments/days devoid of nitrogenous nutrients. However, even without statistical differences in polyphenols and carbohydrate production, the study identifies 5 mM nitrate as the optimal concentration for primarily enhancing total MAAs and the antioxidant activity in *P. linearis*. This discovery highlights the nuanced role of nitrate in algal physiology, suggesting that the overall biological and chemical responses to nitrate supplementation can optimize the organism’s health and its commercially significant bioactive potential for cosmeceutical applications, such as MAAs. These compounds are known for their photoprotective properties; promoting cell proliferation; activating NRF-2, a regulator of cellular oxidative stress; protecting against DNA damage; and exhibiting anti-aging effects [59,70,71,72]; moreover, MAAs are a natural and safe alternative when compared to the use of cytotoxic sunscreens [73]. Furthermore, given its ability to develop and absorb high doses of nitrate, this alga becomes an excellent option for IMTA cultivation and bioremediation.

## Figures and Tables

**Figure 1 marinedrugs-22-00222-f001:**
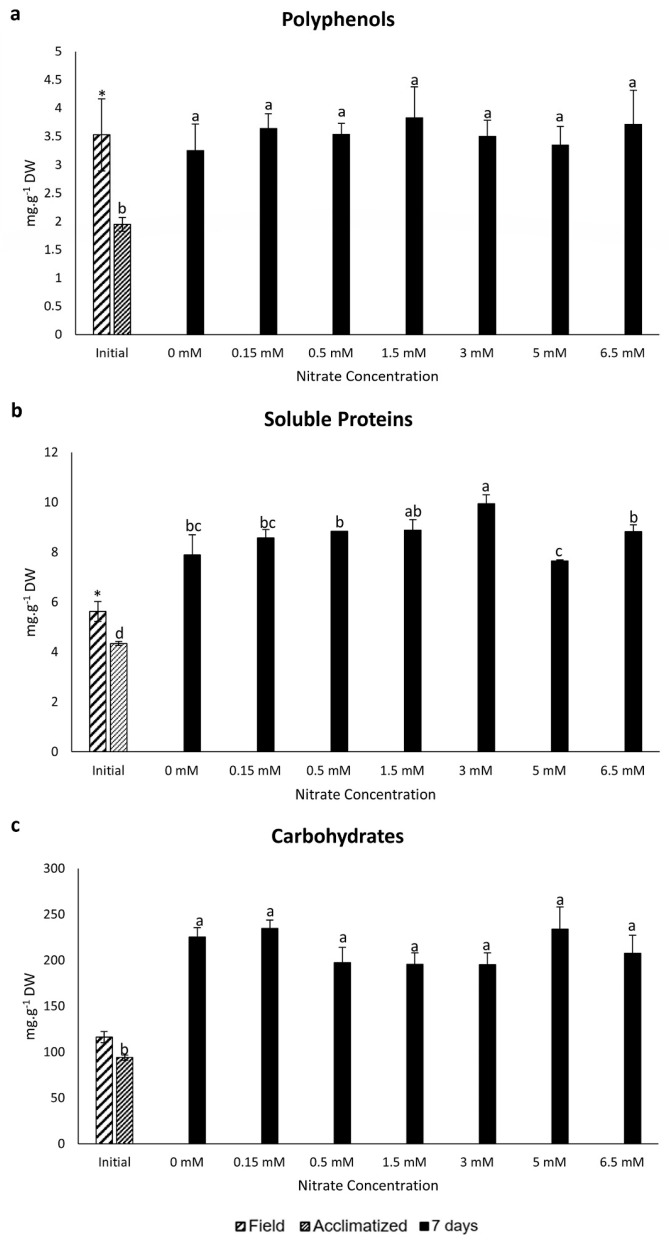
Concentrations of polyphenols (**a**), soluble proteins (**b**), and soluble carbohydrates (**c**) (mg·g^−1^ dry weight) in *P. linearis* after exposure to different nitrate concentrations for seven days (*n* = 3; mean ± SD). Different letters indicate significant differences according to the one-way analysis of variance (considering nitrate concentrations as a fixed factor) and Tukey’s test (*p* ≤ 0.05). Asterisks indicate significant differences according to the independent *t*-test analysis (between initial samples: field and acclimatized) (*p* ≤ 0.05).

**Figure 2 marinedrugs-22-00222-f002:**
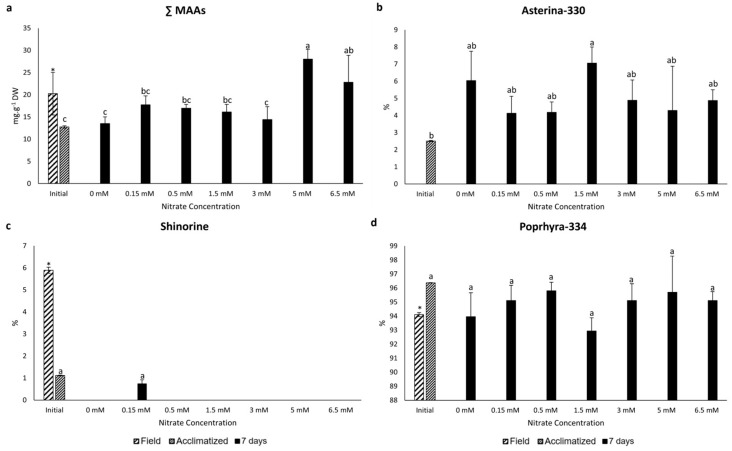
Concentrations total of mycosporine-like amino acids (sum of all identified MAAs) (**a**) (mg·g^−1^ dry weight) and percentage of each MAA (asterina-330 (**b**), shinorine (**c**), and porphyra-334 (**d**)) in *P. linearis* after exposure to different nitrate concentrations for seven days (*n* = 3; mean ± SD). Different letters indicate significant differences according to the one-way analysis of variance (considering nitrate concentrations) and Tukey’s test (*p* ≤ 0.05). The asterisk indicates significant differences according to the independent *t*-test analysis (between initial samples: field and acclimatized) (*p* ≤ 0.05).

**Figure 3 marinedrugs-22-00222-f003:**
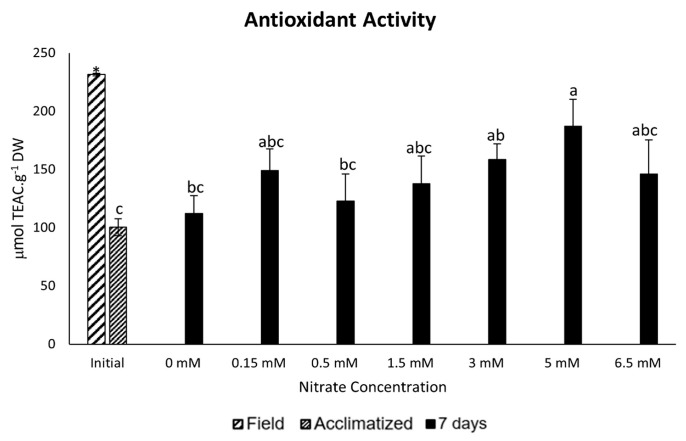
Antioxidant capacity expressed as μmol of Trolox (TEAC)·g^−1^ dry weight in *P. linearis* after exposure to different nitrate concentrations for seven days (*n* = 3; mean ± SD). Different letters indicate significant differences according to the one-way analysis of variance (considering nitrate concentrations) and Tukey’s test (*p* ≤ 0.05). The asterisk indicates significant differences according to the independent *t*-test analysis (between initial samples: field and acclimatized) (*p* ≤ 0.05).

**Figure 4 marinedrugs-22-00222-f004:**
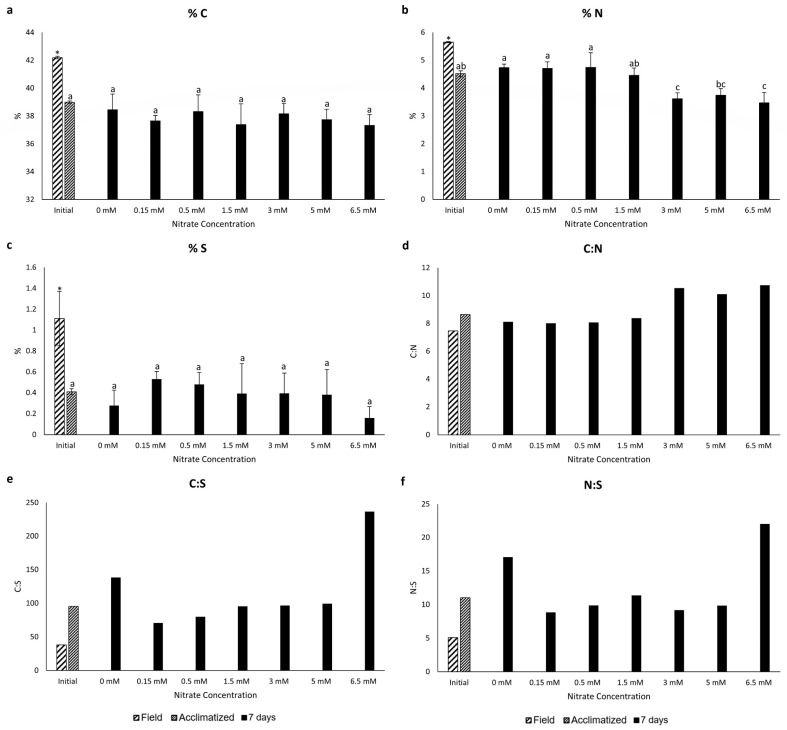
Content of carbon (C) (**a**), nitrogen (N) (**b**), and sulfur (S) (**c**) (% dry weight), and the ratios C:N (**d**), C:S (**e**), and N:S (**f**) in *P. linearis* after exposure to different nitrate concentrations for seven days (*n* = 3; mean ± SD). Different letters indicate significant differences according to the one-way analysis of variance (considering nitrate concentrations) and Tukey’s test (*p* ≤ 0.05). Asterisks indicate significant differences according to the independent *t*-test analysis (between initial samples: field and acclimatized) (*p* ≤ 0.05).

**Figure 5 marinedrugs-22-00222-f005:**
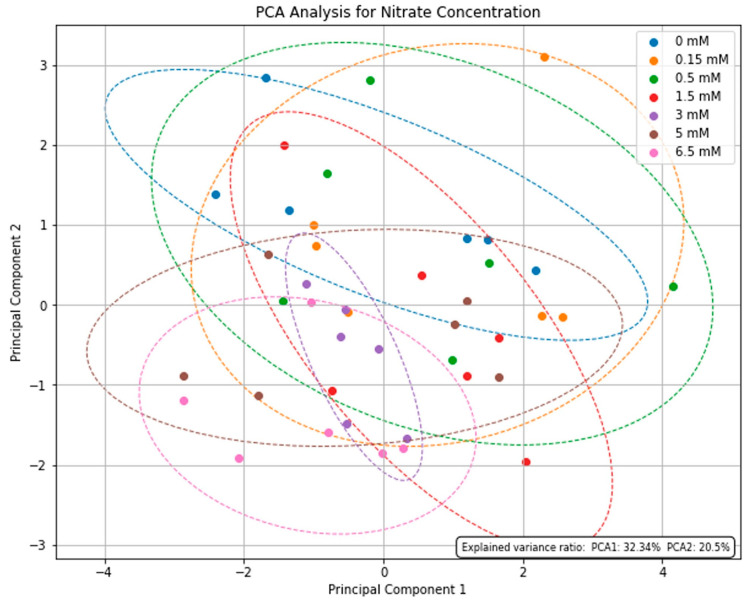
Principal component analysis (PCA) for nitrate concentration of BACs (polyphenols, soluble proteins, carbohydrates, total MAAs), antioxidant activity (ABTS)), and total content of carbon (C), nitrogen (N), and sulfur (S) of *P. linearis*, considering nitrate concentration.

**Table 1 marinedrugs-22-00222-t001:** Nitrate uptake efficiency (NUE, %) and nitrate uptake rate (NUR, mmol of NO_3_^−^·g^−1^ DW·day^−1^) in *P. linearis* in the first two days and in the last five days of exposure to initial nitrate concentrations of 0, 0.15, 0.5, 1.5, 3, 5, and 6.5 mM (*n* = 3; mean ± SD). Different letters indicate significant differences according to the one-way analysis of variance (nitrate concentrations), followed by Tukey’s post hoc test (*p* ≤ 0.05).

Nitrate Concentration	NUE % (0–2 Days)	NUE % (2–7 Days)	NUR mmol·g^−1^ DW·day^−1^ (0–2 Days)	NUR mmol·g^−1^ DW·day^−1^ (2–7 Days)
0 mM	-	-	-	-
0.15 mM	86.25 ± 1.66 ^a^	84.28 ± 2.96 ^ab^	0.11 ± 0.01 ^ab^	0.00 ± 0.00 ^d^
0.5 mM	94.74 ± 2.64 ^a^	89.46 ± 5.15 ^a^	0.42 ± 0.15 ^a^	0.01 ± 0.002 ^d^
1.5 mM	24.38 ± 7.57 ^b^	76.17 ± 6.94 ^b^	0.31 ± 0.11 ^ab^	0.29 ± 0.02 ^c^
3 mM	8.30 ± 5.40 ^c^	49.63 ± 6.09 ^c^	0.20 ± 0.13 ^ab^	0.42 ± 0.05 ^a^
5 mM	6.26 ± 3.79 ^c^	22.78 ± 1.50 ^d^	0.27 ± 0.16 ^ab^	0.37 ± 0.04 ^ab^
6.5 mM	1.75 ± 1.39 ^c^	14.60 ± 1.10 ^d^	0.10 ± 0.08 ^b^	0.33 ± 0.02 ^bc^

## Data Availability

Data are available upon request.

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
