# Peer review of "The Role of Nitrate Supply in Bioactive Compound Synthesis and Antioxidant Activity in the Cultivation of Porphyra linearis (Rhodophyta, Bangiales) for Future Cosmeceutical and Bioremediation Applications"

_marinedrugs, 2024, doi:10.3390/md22050222_

Round 1

Reviewer 1 Report

Comments and Suggestions for Authors

The results presented in the manuscript are of interest and I am in favour of their publication in the journal Marine Drugs, after a minor revision of the manuscript. I note minor because, from my opinion, the authors need to correct some minors things. I would advise the authors to reply to some questions I had during the reading of the manuscript so that they can tell a good story.

Title: Put in italics the genus and species names Porphyra linearis

Cosmeceutical applications… I wonder if this end of the title is correct as no cosmeceutical test were carried out in the study… The end in on Bioremediation… the title could be changed

The aim of the study is to find good conditions to cultivate Porphyra linearis under N supply? Metabolic orientation?

Abstract: Porphyra in italics

Introduction:

The introduction is short

Perhaps a paragraph should be added on the interest of Porphyra in general, as the authors immediately focus on bioremediation.

Porphyra parts are grown all over the world... Where? is it grown in Europe? is it produced worldwide? in Europe?

Add a paragraph on cultivation conditions (basins? open sea?)... are there any enrichments or is the seaweed cultivated as is in the producing countries?

It would be interesting to show the potential of species in the Porphyra genus: food, cosmetics... and help the reader understand why the focus is on MAAs... Porphyra-334 used in cosmetics, etc.

Results:

Figure 1a: the polyphenol contents of the experiment are similar to those of the field batch... Only the acclimatized batch has less. What is the explanation for this?

The results shown in Figure 1 are interesting, but raise questions:

lots of protein compared to field and acclimatised batches

Why, at the initial stage of the experiment, is it not the same as the field or acclimatised batches?

Why does protein levels drop at 5mM?

a lot of carbohydrates compared to field and acclimatised batches

Could you discuss these results?

Figure 2. Why is there such a difference between MAA sum (figure a) and % Maa (figure b)... when the majority is Porphyra-334... so if I understand the calculation method correctly, the results presented in figure 2a and 2b should be similar?

The field sample has shinorin, very little shinorin is found in the acclimatised sample and none at all in the cultured samples (under N supply... apart from 0.15 mM where very small quantities, a bit like acclimatised samples).

Could the authors explain why?

Is there an explanation for this?

Can we say that the acclimatised batch was grown with 0.15 mM N, since shinorin is present in both batches: acclimatised and with 0.15 mM?

Materials and methods

It would be interesting to know what the level of N was in the seawater in the area where the samples were taken.

As the title mentions cosmeceutical applications… other tests could have been used to support the cosmeceutical potential... cosmetic tests, ORAC test, etc...

Conclusion

how do the authors envisage growing Porphyra linearis? in in-door/out-door ponds? in situ, especially in eutrophic zones?

In the abstract, similar end sentence as in this section conclusion…

Authors did not develop the idea of bioremediation in the Discussion section… It will be nice to developpe a little more to understand the last sentence. Then, authors could add a paragraph in the Discussion on the potential of Porphyra linearis in bioremediation, in compare its potential with others species

Author Response

Comments and Suggestions for Authors

The results presented in the manuscript are of interest and I am in favour of their publication in the journal Marine Drugs, after a minor revision of the manuscript. I note minor because, from my opinion, the authors need to correct some minors things. I would advise the authors to reply to some questions I had during the reading of the manuscript so that they can tell a good story.

Title: Put in italics the genus and species names Porphyra linearis

Cosmeceutical applications… I wonder if this end of the title is correct as no cosmeceutical test were carried out in the study… The end in on Bioremediation… the title could be changed

The aim of the study is to find good conditions to cultivate Porphyra linearis under N supply? Metabolic orientation?

Response: We agree, the main goal of the present study was to find the best concentration of nitrate in water to enhance the production of bioactive compounds (BACs), which can be used in cosmeceuticals. We made the appropriate changes in the title.

Abstract: Porphyra in italics

Response: We agree, and we made the appropriate changes in the text.

Introduction:

The introduction is short

Perhaps a paragraph should be added on the interest of Porphyra in general, as the authors immediately focus on bioremediation.

Porphyra parts are grown all over the world... Where? is it grown in Europe? is it produced worldwide? in Europe?

Add a paragraph on cultivation conditions (basins? open sea?)... are there any enrichments or is the seaweed cultivated as is in the producing countries?

It would be interesting to show the potential of species in the Porphyra genus: food, cosmetics... and help the reader understand why the focus is on MAAs... Porphyra-334 used in cosmetics, etc.

Response: We agree, and we have added information in the introduction.

Results:

Figure 1a: the polyphenol contents of the experiment are similar to those of the field batch... Only the acclimatized batch has less. What is the explanation for this?

Response: We agree, and as described in the discussion of the article: ‘In plants and algae cultivated under stress conditions, such as nitrogen deficiency, a redirection of carbon reserves from secondary metabolism towards biomass accumulation can occur. This can explain the decrease in polyphenol production in the absence of nitrate that was observed in acclimatized samples. Furthermore, even though the acclimatization period was conducted under the same conditions and density as the experiment, the acclimatization took place in tanks, and at times the thalli became more shaded, and the reduction in of absorbed light, combined with the absence of nutrients, may have contributed to decreasing the quantity of the compounds, such as polyphenols. After acclimation and setting up the experiment, each replicate was cultivated in cylinders, preventing shading, which may have favored ETR and facilitated the increase in metabolites.’

The results shown in Figure 1 are interesting, but raise questions:

lots of protein compared to field and acclimatised batches.  

Why, at the initial stage of the experiment, is it not the same as the field or acclimatised batches?

Response: We agree, and as described in the discussion of the article: ‘Even in the absence of nitrate, the concentration of soluble proteins increased, which may be associated with more light intensity and the degradation of other cellular structures to produce these defensive antioxidant compounds.’

Why does protein levels drop at 5mM?

Response: We agree, and we wrote a possible explanation for this decline in the paragraph discussing MAAs. It is precisely in the 5 mM treatment that we have a low concentration of soluble proteins but high production of MAAs, suggesting a possible shift in the nitrogen utilization pathway in this treatment.

a lot of carbohydrates compared to field and acclimatised batches

Could you discuss these results?

Response: We agree, and we have added some explanation for these results in the discussion.

Figure 2. Why is there such a difference between MAA sum (figure a) and % Maa (figure b)... when the majority is Porphyra-334... so if I understand the calculation method correctly, the results presented in figure 2a and 2b should be similar?

Response:  Figure ‘a’ represents the total MAA (sum of all MAAs types detected) in mg per gram of dry biomass. Figure ‘b’ calculates the percentage of each type of identified MAA for each treatment. In other words, the total for each treatment presented in Figure ‘a’ represents 100%, and from that 100%, the percentage of each type of MAA is calculated. But, we agree that the image is difficult to interpret and have changed it to separate graphs (Fig. 2 new).

The field sample has shinorin, very little shinorin is found in the acclimatised sample and none at all in the cultured samples (under N supply... apart from 0.15 mM where very small quantities, a bit like acclimatised samples).

Could the authors explain why?

Is there an explanation for this?

Can we say that the acclimatised batch was grown with 0.15 mM N, since shinorin is present in both batches: acclimatised and with 0.15 mM?

Response: The acclimation was carried out with 0 mM of nitrate. In this situation, what we observe is that the period in laboratory culture causes the amount of shinorine to decrease drastically. It is noted that the percentage of Porphyra-334 does not change statistically with the treatments, therefore there is a substitution of shinorine for asterina-330, as described by Peng et al. 2023. Therefore, in these thalli exposed to 0.15 mM, this transition has not yet been completed, but we expect it to occur within a few more days of cultivation. The visualization becomes clearer with the new figure inserted (Fig. 2 new). We have added this explanation in the discussion text.

Materials and methods

It would be interesting to know what the level of N was in the seawater in the area where the samples were taken.

Response: We agree, it is an interesting parameter. In Galicia, Spain, the average nitrate level in seawater during winter is 7 µM (data taken from the 'My Ocean Pro - Copernicus Marine Open Data' platform). It is important to consider that this nutrient is constantly renewed through the decomposition of organisms, the upwelling of nutrient-rich deep waters, and the excretion of ammonia by fish, invertebrates, and marine vertebrates, which can be transformed into nitrate by bacteria. We have added this information in material and methods text.

As the title mentions cosmeceutical applications… other tests could have been used to support the cosmeceutical potential... cosmetic tests, ORAC test, etc...

Response: We agree with you; however, we did not conduct cosmeceutical tests in this study as they are already described in the literature (Vega et al., 2021; Chrapusta e tal., 2017; Figueroa, 2021; Singh et al., 2021; Peng et al., 2023 - cited in the conclusion). The aim of this research is to identify the best nitrate conditions to enhance the production of bioactive compounds with cosmeceutical applications.

Conclusion

how do the authors envisage growing Porphyra linearis? in in-door/out-door ponds? in situ, especially in eutrophic zones?

Response: Porphyra sp. is a very resilient algae, capable of being cultivated in eutrophic zones, or even in out- and in-door tanks with the addition of nitrate to the water. Furthermore, the next steps for the research group are to cultivate it with high levels of nitrate, derivate form fishponds, along with ultraviolet radiation to test if it is possible to further increase the production of bioactive compounds of cosmeceutical interest. We have added some this information in the conclusion text.

In the abstract, similar end sentence as in this section conclusion…

Authors did not develop the idea of bioremediation in the Discussion section… It will be nice to developpe a little more to understand the last sentence. Then, authors could add a paragraph in the Discussion on the potential of Porphyra linearis in bioremediation, in compare its potential with others species

Response:  We agree, and we have added a phrase in the abstract.

Reviewer 2 Report

Comments and Suggestions for Authors

The study addresses an important topic of optimizing the cultivation conditions for enhancing the production of valuable bioactive compounds from the red macroalga Porphyra linearis. The experimental design is well-described, and a range of nitrate concentrations were tested. Multiple bioactive compounds were analyzed, including polyphenols, proteins, carbohydrates, mycosporine-like amino acids (MAAs), and antioxidant activity. The authors have related their findings to the potential applications in cosmeceuticals, integrated multi-trophic aquaculture (IMTA), and bioremediation. Appropriate statistical analyses were conducted, and the results were presented clearly with figures and tables. While there are some minor comments that need to be addressed to improve this MS.

1.        The introduction could be improved by focusing on the specific research question and objectives.

2.        It does seem illogical for the 0 mM nitrate treatment samples to have higher levels of certain bioactive compounds (BACs) compared to the acclimatized samples, given that the acclimatized samples were used as the starting material for the nitrate experiment.

3.        The discussion section could be expanded to provide more in-depth interpretation of the results and their implications, as well as comparisons with relevant literature.

4.        The authors should discuss the potential mechanisms by which nitrate influenced the biosynthesis of different bioactive compounds, especially MAAs and antioxidants.

5.        Some sections of the manuscript, particularly the results and discussion, could benefit from more rigorous editing to improve the clarity and flow of the text.

Overall, the study presents valuable findings on the optimization of nitrate concentrations for enhancing the production of bioactive compounds in P. linearis. With some additional discussion and elaboration, as well as addressing the weaknesses mentioned above, the manuscript could potentially be suitable for publication in a peer-reviewed journal in the field of marine drugs or algal biotechnology.

Author Response

Comments and Suggestions for Authors

The study addresses an important topic of optimizing the cultivation conditions for enhancing the production of valuable bioactive compounds from the red macroalga Porphyra linearis. The experimental design is well-described, and a range of nitrate concentrations were tested. Multiple bioactive compounds were analyzed, including polyphenols, proteins, carbohydrates, mycosporine-like amino acids (MAAs), and antioxidant activity. The authors have related their findings to the potential applications in cosmeceuticals, integrated multi-trophic aquaculture (IMTA), and bioremediation. Appropriate statistical analyses were conducted, and the results were presented clearly with figures and tables. While there are some minor comments that need to be addressed to improve this MS.

  1. The introduction could be improved by focusing on the specific research question and objectives.

Response: We agree, and we have added information in the introduction.

  1. It does seem illogical for the 0 mM nitrate treatment samples to have higher levels of certain bioactive compounds (BACs) compared to the acclimatized samples, given that the acclimatized samples were used as the starting material for the nitrate experiment.

Response:  We agree, and we wrote sometimes in the discussion part that even in the absence of nitrate, the concentration of some BACs increased, which may be associated with more light intensity and the degradation of other cellular structures to produce these compounds.

  1. The discussion section could be expanded to provide more in-depth interpretation of the results and their implications, as well as comparisons with relevant literature.

Response:  We agree, and we have added information in the discussion and new references.

  1. The authors should discuss the potential mechanisms by which nitrate influenced the biosynthesis of different bioactive compounds, especially MAAs and antioxidants.

Response:  We agree, and we have added information in the discussion about the production pathways of MAAs and phenolics, the main antioxidant BACs with cosmeceutical properties.

  1. Some sections of the manuscript, particularly the results and discussion, could benefit from more rigorous editing to improve the clarity and flow of the text.

Response:  We agree, and we made some changes to the text to make it more fluid.

Overall, the study presents valuable findings on the optimization of nitrate concentrations for enhancing the production of bioactive compounds in P. linearis. With some additional discussion and elaboration, as well as addressing the weaknesses mentioned above, the manuscript could potentially be suitable for publication in a peer-reviewed journal in the field of marine drugs or algal biotechnology.
